# Effects on Metabolism in Astrocytes Caused by cGAMP, Which Imitates the Initial Stage of Brain Metastasis

**DOI:** 10.3390/ijms22169028

**Published:** 2021-08-21

**Authors:** Toya Okawa, Kurumi Hara, Momoko Goto, Moe Kikuchi, Masataka Kogane, Hiroto Hatakeyama, Hiroki Tanaka, Daiki Shirane, Hidetaka Akita, Akihiro Hisaka, Hiromi Sato

**Affiliations:** 1Clinical Pharmacology and Pharmacometrics, Graduate School of Pharmaceutical Sciences, Chiba University, 1-8-1 Inohana, Chuo-ku, Chiba-shi, Chiba 260-8675, Japan; afna6651@chiba-u.jp (T.O.); kuruminyanx3ko@gmail.com (K.H.); peachko_27.vb_j@icloud.com (M.G.); rampianko@gmail.com (M.K.); afsa6614@chiba-u.jp (M.K.); h-hatakeyama@chiba-u.jp (H.H.); hisaka@chiba-u.jp (A.H.); 2Design and Drug Disposition, Graduate School of Pharmaceutical Sciences, Chiba University, 1-8-1 Inohana, Chuo-ku, Chiba-shi, Chiba 260-8675, Japan; hiroki_tanaka8922@chiba-u.jp (H.T.); ds.white.89@gmail.com (D.S.); akitahide@chiba-u.jp (H.A.)

**Keywords:** cGAMP, metabolome shift, astrocyte, metastatic brain tumor, glutamate

## Abstract

The second messenger 2′3′-cyclic-GMP-AMP (cGAMP) is thought to be transmitted from brain carcinomas to astrocytes via gap junctions, which functions to promote metastasis in the brain parenchyma. In the current study, we established a method to introduce cGAMP into astrocytes, which simulates the state of astrocytes that have been invaded by cGAMP around tumors. Astrocytes incorporating cGAMP were analyzed by metabolomics, which demonstrated that cGAMP increased glutamate production and astrocyte secretion. The same trend was observed for γ-aminobutyric acid (GABA). Conversely, glutamine production and secretion were decreased by cGAMP treatment. Due to the fundamental role of astrocytes in regulation of the glutamine–glutamate cycle, such metabolic changes may represent a potential mechanism and therapeutic target for alteration of the central nervous system (CNS) environment and the malignant transformation of brain carcinomas.

## 1. Introduction

Astrocytes express a variety of transporters and are responsible for central nervous system (CNS) homeostasis and neuronal protection by taking up or releasing numerous substances [1]. Glucose in the blood is taken up by astrocytes mainly through glucose transporter 1 (GLUT1), which is expressed in endothelial cell walls of capillary vessels and astrocyte membranes. It has been proposed that astrocytes metabolize and release glucose to lactate, which is then taken up by neurons and serves as a substrate for the tricarboxylic acid (TCA) cycle (astrocyte–neuron lactate shuttle (ANLS)) [2,3]. Astrocytes are the only cells in the brain that accumulate abundant glycogen granules and can also produce lactic acid through the breakdown of glycogen [4]. Astrocytic processes cover presynapses and postsynapses to form tripartite synapses, which communicate closely with neurons in both directions [5]. Astrocytes collect glutamate (Glu) released from presynapses to improve the efficiency of synaptic transmission and prevent neurotoxicity due to excess extracellular Glu [6]. Astrocytes convert the captured Glu into glutamine (Gln) by glutamine synthetase (GS) and subsequently release it [7]; neurons then take the released Gln up, convert it to glutamate, and reuse it. This sequence is called the Gln-Glu cycle [8]. Astrocytes can also produce Glu via an amino-transfer reaction from branched-chain amino acids (BCAA) to α-ketoglutarate, an intermediate in the TCA cycle, and convert it to glutamine for supply to neurons. In addition, astrocytes increase intracellular Ca^2+^ concentrations in response to synaptic activity. This increase in concentration propagates as a Ca^2+^ wave to surrounding astrocytes, causing the release of gliotransmitters such as ATP, Glu, D-serine, and GABA, which are also neurotransmitters and affect the activity of neural circuits [8].

Astrocytes are known to change their properties under pathophysiological conditions, such as trauma, infection, ischemia, and degenerative diseases of the CNS [9]. These astrocytes are called reactive astrocytes, which generally show a thickened morphology with increased expression of glial fibrillary acidic protein (GFAP). Given that astrocytes actively communicate with neurons, changes in their properties have a significant impact on the brain. However, the nature of reactive astrocytes is diverse and controversial, as they have been reported to either improve or worsen the condition [9,10]. In the context of brain tumors, changes in the properties of the tumor surrounding astrocytes have not been fully elucidated, but it has been reported that at least two types of substances, Ca^2+^ and the cyclic dinucleotide, 2′3′-cyclic-GMP-AMP (cGAMP), are exchanged through gap junctions (GJs) formed between cancer cells and astrocytes. Although elevated intracellular Ca^2+^ induces apoptosis, GJs with astrocytes reduce excessive Ca^2+^ in melanoma cells induced by anticancer drugs [11]. In addition, cGAMP is transmitted from brain-metastatic breast and lung cancer cells through GJs into astrocytes, promoting brain metastasis [12].

cGAMP is produced by cGAMP synthase (cGAS) from cytosolic dsDNA and serves as a stimulator of interferon genes (STING), and this cGAS-STING pathway is of particular interest in the field of tumor immunity. The cGAS-STING pathway transmits innate immune response signals to cytoplasmic DNA [13]. STING, an adaptor molecule on the endoplasmic reticulum, activates downstream signaling molecules such as TBK1, IRF3, IKK, and NF-κB to induce type 1 interferons (IFNs) and inflammatory cytokines [14]. In cancer pathogenesis, DNA leaks from the nucleus due to rapid proliferation and is present in the cytoplasm, which serves as a substrate for cGAS to produce cGAMP. When this reaches noncancerous cells via a transduction pathway, it stimulates the STING pathway. When acting on the STING pathway in dendritic cells, the produced type 1 IFNs mature dendritic cells and drive a tumor immune response by stimulating cross-presentation to CD8+ T cells [15]. It has also been reported that tumor-derived cGAMPs stimulate immune responses by NK cells [16]. Therefore, drugs that activate the STING pathway have attracted attention as a new cancer treatment strategy, and cancer cells have been reported to regress after the administration of STING ligands [17,18,19,20]. However, it should be noted that activation of the cGAS-STING pathway does not always exert an antitumor effect and can be a double-edged sword in cancer treatment [21,22]. Cancer cells are known to resist activation of the cGAS-STING pathway, and it has been reported that silencing of STING or cGAS prevents the induction of antitumor immune responses in colorectal cancer [23], malignant melanoma [24], ovarian cancer [25], and KRAS-mutant non-small-cell lung cancer (NSCLC) [26]. It is also known that STING-mediated inflammation can promote the development and proliferation of cancer [27,28].

The central metabolic regulation of astrocytes makes the interstitial space of the brain a unique metabolic environment. Indeed, it has been reported that cancer cells that metastasize to the brain parenchyma may reprogram their metabolism to survive and, consequently, exhibit characteristics different from those of the primary site [29,30,31]. Recently, we reported that tumor and astrocyte coculture systems may undergo metabolic changes in each other prior to phenotype changes [32]. However, the involvement of cGAMP in this context is unclear. Based on the hypothesis that cGAMP alters central metabolic regulation, an important physiological function of astrocytes, and contributes to the formation of the early-stage cancer microenvironment, we aimed to evaluate the effects of cGAMP on astrocyte metabolism. Because cGAMP is highly hydrophilic, we used a special lipid nanoparticle called ss-cleavable pH-activated lipid-like material (ssPalm) to deliver cGAMP into astrocytes. It is a functional molecule with three units: a lipophilic scaffold for membrane structure formation, a tertiary amine for pH sensitivity, and a disulfide bond for intracellular disintegration [33,34,35,36,37]. The carrier surface formed by tertiary amines is neutral under physiological pH environments and avoids electrostatic nonspecific interactions, while it facilitates escape from the endosome by being charged in a low pH environment in the endosome [37,38]. In addition, disulfide bonds can be cleaved in the intracellular reducing environment, causing destabilization of the membrane structure and allowing active release of nucleic acids [36,37]. It is the first attempt to introduce cGAMP via ssPalm. It was finally compared with the amount of intracellular cGAMP and metabolome changes in cGAMP-astrocytes to those in the control group.

## 2. Results

### 2.1. Delivery of cGAMP to Astrocytes

As cGAMP is an electrically charged molecule, it cannot pass through the cell membrane of astrocytes in its original state. Therefore, we attempted intracellular delivery of cGAMP using lipid nanoparticles based on ssPalm. To determine optimal formulation in the preliminary study, dioleoyl-sn-glycero-phosphatidylcholine (DOPC) or dioleoyl-sn-glycero-phosphoethanolamine (DOPE) was used as the lipid component. The cGAMP/lipid ratio was tried for 50 nmoL/µg and 200 nmoL/µg (Appendix A). Under these conditions, particle properties were evaluated by dynamic light scattering. The particle size and the polydispersity index (PdI) were not significantly different among the samples (Appendix A). The comparison of the efficiency of cGAMP delivery revealed that the particle with DOPE and smaller cGAMP/lipid ratio (50 nmoL/µg) demonstrated relatively higher delivery efficiency of cGAMP into the astrocytes (Appendix A). Because the incorporation of phosphoethanolamine enhances the endosomal escape efficiency of lipid nanoparticles, intracellular trafficking was improved by the presence of the DOPE. Therefore, the composition was employed for the following experiments.

The ssPalm particles encapsulating cGAMP (ssPalm-cGAMP) were taken up by astrocytes, and intracellular cGAMP levels were examined (Figure 1A). The results showed that the amount of cGAMP was below the detection limit in the astrocytes that took up nanoparticles without cGAMP (ssPalm-empty), but that cGAMP was detected in the astrocytes that took up ssPalm-cGAMP.

The amount of cGAMP in astrocytes of the ssPalm-cGAMP group was comparable to that in the cells of the human triple-negative breast cancer cell line, MDA-MB-231 (MDA231). As expected, a significant increase in the amount of cGAMP was observed in MDA231 after DNA damage by doxorubicin (Dox). When other types of cancers with the potential to metastasize to the brain were also examined, significantly lower amounts of cGAMP were detected in A549 lung cancer and DLD-1 and HCT-15 colorectal cancer compared to MDA231. There was also no significant increase in cGAMP by Dox in those cancer cells (Figure 1A).

We then examined the secretion of IFN-β, a cytokine induced downstream of STING, to confirm the responsiveness of the STING pathway of astrocytes to externally introduced cGAMP (Figure 1B). The results showed significantly increased IFN-β in the culture supernatant of astrocytes that had taken up ssPalm-cGAMP. The mRNA expression of cytokines induced downstream of STING, *Ifnb1* and *Tnf*, was also significantly increased in the ssPalm-cGAMP group (Figure 1C). These results confirm that ssPalm particles can deliver cGAMP into astrocytes and that STING signaling is responsive in astrocytes.

### 2.2. Alteration of Glucose Metabolism by cGAMP

The transfer of cGAMP from cancer cells may contribute to the formation of the cancer microenvironment by altering the metabolic function of astrocytes. Glucose and glutamate metabolism play important roles in the central metabolic regulation of astrocytes. Therefore, we investigated cGAMP-induced metabolic changes in astrocytes by focusing on these two pathways. 

By incorporating ^13^C6-glucose into astrocytes as a tracer, the changes that occur in glucose-related metabolism were quantified by labeled metabolites (Figure 2B and Figure 3A–C). One hour after the addition of ^13^C6-glucose, most of the metabolites of glycolysis (C6, C3) were replaced by ^13^C isotopomers (Figure 2B, glycolysis), but lactic acid (lactate) was only partially replaced after 60 min due to its overwhelmingly intracellular metabolites (Figure 2B). The only glycolytic metabolite detected in the supernatant was the ^13^C isotopomer of lactate (Figure 2B, bottom right). Although there were not many ^13^C substituents other than glycolysis, R5P and S7P were identified in the metabolites of the pentose phosphate pathway (PPP), and malic acid and acetyl-CoA in the TCA cycle (Figure 2B, PPP and TCA cycle). On the other hand, based on the ^13^C isotopomer occupancy of malate and acetyl-CoA, almost half of the metabolites related to glucose metabolism were considered to be substituted.

To compare the empty and cGAMP groups, metabolomic analysis was performed using the concentration information with the highest percentage of ^13^C substituents for each metabolite detected (Appendix A). In the volcano plot, which was analyzed by normalizing the concentrations, Ribose-5-phosphate (R5P) was the only compound that showed a significant increase of more than 1.5-fold in the cGAMP group (Figure 2D). In the pathway analysis conducted to interpret the results, besides PPP, other pathways such as lipid metabolism, pyruvate metabolism, glycolysis/glycogenesis, and TCA cycle were included in the top 10 (Figure 2E, Appendix A). Noting the variation in the ^13^C substitution ratio, these changes should be investigated in the future.

To confirm the effect on the first step of glucose uptake in cells, the amount of 2-deoxy glucose-6-phosphate (2DG6P) in cells was quantified by enzymatic cycling 1 h after the addition of 2-deoxy glucose (2DG); 2DG taken up into the cells accumulates as 2DG6P without undergoing subsequent metabolism, and the amount indicates the glucose uptake capacity. As a result, the intracellular 2DG6P level was significantly increased by approximately 1.1-fold with cGAMP compared to that in the empty cells (Figure 2C).

Regarding the metabolism of Glu and Gln, the amino acids produced by the TCA cycle, the substitution rate to ^13^C-labeled compounds after one hour of addition of ^13^C6-glucose was about half for Glu (Figure 3A, Glu stacked graph) and one quarter for Gln (Figure 3B, Gln stacked graph). The intracellular level of ^13^C2-Glu, which is the most abundant ^13^C-labeled isotopomer (Figure 3A, Glu stacked graph), was significantly increased in the cGAMP group (Figure 3A, 13C2), while that of ^13^C2-Gln was significantly decreased in the cGAMP group (Figure 3B, 13C2). In addition, ^13^C0-Gln showed a decreasing trend (Figure 3B, 13C0), but ^13^C0-Glu showed an increasing trend (Figure 3A, 13C0). The unlabeled ^13^C0 isotopomer was present before labeling, and in the case of Glu, the net result of the direct source of Glu from α-ketoglutarate and glutamine in the TCA cycle, as well as what was consumed, is the current amount. Considering this, the present results suggest that the conversion of Gln to Glu may be accelerated. The substitution rate of the TCA cycle estimated from malic acid was about half (Figure 2B, TCA cycle), and the fact that the ^13^C isotopomer occupancy of Glu is almost equal to this (Figure 3A, Glu stacked graph), while it is less for Gln (Figure 3B, Gln stacked graph), also supports this hypothesis.

Furthermore, the amount of Gln released into the culture supernatant was lower in the cGAMP group than in the empty group for all labeled Gln and ^13^C0-Gln (Figure 3C). Extracellular Glu was examined in the same way, but the amount in the supernatant was below the detection limit (data not shown).

### 2.3. Alteration of Glutamine–Glutamate (Gln-Glu) Metabolism by cGAMP

The effect of cGAMP on Gln-Glu metabolism in astrocytes was assessed by incorporating ^13^C5,^15^N-Glu as a tracer into cGAMP-transfected astrocytes and quantifying the labeled metabolites (Figure 4A–D). The substitution of intracellular Glu and Gln for labeled ones progressed with time from 5 min to 40 min after the addition of ^13^C5,^15^N-Glu (Figure 4B,C, stacked graphs). For intracellular Glu, the amount of unlabeled M+0 decreased in a time-dependent manner, while those of M+6, which incorporated ^13^C5,^15^N-Glu, and other labeled Glu (M+1, M+5), first increased in a time-dependent manner, and then decreased or reached a plateau (Figure 4B, line graphs). The amount of Glu was higher in the cGAMP group than in the empty group for all labeled Glu. The difference was especially pronounced in the M+6 group, where cGAMP significantly increased the amount of Glu at 5, 10, and 20 min after the addition of ^13^C5,^15^N-Glu compared to the empty group. However, for Gln synthesized from Glu, this relationship was reversed; the intracellular Gln level was lower in the cGAMP group than in the empty group for all labeled Gln isotopomers (Figure 4C, line graphs). In particular, M+5 and M+6 were significantly reduced in the cGAMP group at all time points, 5, 10, 20, and 40 min.

The amount of Glu in the culture supernatant after the addition of ^13^C5,^15^N-Glu was also examined, and the results indicated that the amount of M+0- and M+6-labeled Glu in the cGAMP group was significantly higher than that in the empty group after 40 min (Figure 4D). This result indicates that cGAMP increased the release of Glu, as M+0 in the supernatant was due to the release of Glu from cells that had been stored before the addition of ^13^C5,^15^N-Glu. As for the amount of Gln in the supernatant, the amount of M+6-labeled Gln, which is directly converted from ^13^C5,^15^N-Glu, increased in a time-dependent manner, but there was no significant change by cGAMP.

Given that the amounts of intracellular Glu and Gln were altered by cGAMP, we examined the mRNA expression of glutamine synthetase (GS), an enzyme that produces Gln from Glu, and glutaminase (GLS), an enzyme that produces Glu from Gln (Figure 4E). In contrast to the finding that intracellular Glu was upregulated by cGAMP, mRNA expression of GLS was downregulated by cGAMP, and no change in mRNA expression of GS was observed.

### 2.4. Effect of cGAMP on GABA Production

GABA is a representative amino acid that acts as an inhibitory neurotransmitter in the CNS. The cGAMP-induced increase in intracellular Glu led us to investigate the intracellular levels of GABA, a metabolite directly synthesized from Glu. We added ^13^C6-glucose to astrocytes and followed the newly produced GABA. cGAMP significantly increased the intracellular ^13^C2-labeled GABA produced by the direct synthesis of ^13^C2-Glu (Figure 5A, 13C2).

When ^13^C5,^15^N-Glu was added to the supernatant, M+5-labeled GABA, which was directly synthesized from ^13^C5,^15^N-Glu, showed an upward trend in the cells (Figure 5B, stacked graph, line graph M+5). Furthermore, the amount of unlabeled GABA in the supernatant 40 min after the addition of ^13^C5,^15^N-Glu was significantly increased in the cGAMP group (Figure 5C). The mRNA expression level of GAD1, an enzyme that produces GABA from Glu, was unexpectedly decreased by cGAMP (Figure 5D).

## 3. Discussion

Intracellular cGAMP was detected after the addition of ssPalm-cGAMP, indicating that ssPalm-cGAMP can deliver cGAMP into primary cultured astrocytes. In this case, two factors that possibly affect the amount of cGAMP introduced into cells are the encapsulation efficiency of cGAMP into lipid nanoparticles and the efficiency of delivery of the cGAMP-ssPalm complex into cells. ssPalm and cGAMP are thought to form a complex by electrostatic interaction in *t*-butanol; however, the molecular weight and negative charge of cGAMP are smaller than those of siRNA and mRNA, which were originally expected to be introduced into cells using ssPalm. Therefore, the encapsulation efficiency of cGAMP into ssPalm particles is expected to be lower than that of siRNA and mRNA. Because the separation of encapsulated/free cGAMP from the samples had technical difficulties, the encapsulation efficiency was not calculated. 

The presence of cGAMP in cancer cells was confirmed as expected in MDA231, and the increase in DNA leakage from the nucleus by Dox treatment was considered to increase cGAMP production by cGAS. The amount of cGAMP was close to the lower limit of detection in other cancer cell lines in this study, and the response to Dox was also low. Because cGAS expression varies among cancer cells [39], it will be important to identify cancer types that potentially activate the cGAS-STING pathway, such as MDA231, in order to consider the possibility of brain metastasis.

Next, the metabolic functions of cGAMP-transfected astrocytes were examined, focusing on the glucose and Gln-Glu metabolism. Glucose in the capillaries is mainly taken up by astrocytes via GLUT1 and then stored as glycogen, where glycolysis is involved in producing lactate as an energy source for neurons and used as a substrate for the TCA cycle. The results of follow-up after ^13^C6-glucose uptake demonstrated that cGAMP did not significantly alter the amount of lactate but significantly increased R5P in PPP, suggesting enhanced PPP. Because PPP is involved in the supply of NADPH, which regulates redox, and nucleotide synthesis, it is significant to investigate the effects of cGAMP on these processes. Other metabolites of glucose metabolism were not altered by cGAMP, but 2-DG uptake was significantly increased (Figure 2C). To clarify where cGAMP affects the glucose metabolism, it may be useful to determine parameters in the rate dimension, such as flux, in addition to comparing snapshot concentrations. It has been reported that glucose uptake is increased in reactive astrocytes identified during cerebral ischemia [40,41], indicating that similar properties may occur in astrocytes in contact with tumors. Because PPP enhances metabolic flux [42], which may lead to astrocyte cell growth as gliosis. Gliosis is often observed in diseases of the CNS, including brain tumors. Once it becomes a mature glial scar, it persists for a long time and acts as a barrier to protect not only neural axon regeneration but also inflammatory cells, infectious agents, and non-CNS cells such as cancer cells [43]. Whether cGAMP contributes to the formation of such a microenvironment needs to be confirmed in the future.

Regarding Glu metabolism, cGAMP increased ^13^C2-Glu in cells after the addition of ^13^C6-Glucose (Figure 3A). ^13^C2-Glu is produced when ^13^C3-pyruvate from ^13^C6-glucose is converted to ^13^C2-acetyl-CoA by pyruvate dehydrogenase (PDH) and enters the TCA cycle, it is converted to ^13^C3-oxaloacetate by pyruvate carboxylase (PC) and enters the TCA cycle, or it is converted directly from 2-oxoglutarate to Glu. PC is expressed specifically in astrocytes in the brain and is an important enzyme for Glu synthesis [44]. It should be noted that it was not possible to distinguish whether ^13^C2-Glu was synthesized via PDH or PC by CE-TOFMS evaluation in this study. ^13^C3-Glu, ^13^C4-Glu, and ^13^C5-Glu can occur when they are converted from the second or later rounds of the TCA cycle, or when ^13^C2-Acetyl-CoA-derived ^13^C is added to ^13^C3-oxaloacetate. In the present study, the amount of labeled Glu increased after the addition of ^13^C6-glucose, indicating that the synthesis of Glu by cataplerosis from the TCA cycle was increased. In addition, intracellular Gln levels were decreased despite the increase in the intracellular Glu pool by cGAMP (Figure 3B). These results suggest that cGAMP inhibits Gln synthesis from Glu by GS. The same trend was observed in the evaluation of the change in the amount of Glu-related metabolites after the addition of ^13^C5,^15^N-Glu, in which cGAMP increased the intracellular Glu pool but decreased the conversion to Gln (Figure 4B,C).

GS is the only enzyme in the brain that can convert Glu and ammonia to Gln and is expressed almost exclusively in astrocytes [45]. As the synthesis of Glu and GABA in neurons requires a continuous supply of Gln, Gln synthesis in astrocytes is an important component of the Gln-Glu cycle [46]. For this cycle, GS, GLS, and GAD take significant roles in the synthesis of Gln, Glu, and GABA, respectively. Indeed, it has been reported that alterations in GS expression in astrocytes contribute to the pathophysiology of medial temporal lobe epilepsy (MTLE), in which reactive astrocytes are deficient in GS [47,48]. In the present study, cGAMP did not affect the expression level of *Gs*; further, *Gls* and *Gad1* were rather decreased, but the enzyme activities still need to be investigated. 

The decrease in intracellular Gln may be the result of both suppressed conversion of Glu to Gln and the accelerated consumption of Gln, which is a nitrogen source in biosynthesis and is used for amino acid and purine synthesis. It is also consistent with the possibility that cGAMP may promote nucleotide synthesis, as seen in the increase in R5P. We have not yet been able to follow and evaluate the intracellular metabolism of labeled Gln in astrocytes. It has been reported that Gln is actively consumed for purine synthesis in gliomas [49], which is also an interesting target for cGAMP stimulation to investigate.

Following the addition of ^13^C5,^15^N-Glu, the M+0-labeled Glu in the supernatant was increased by cGAMP, suggesting that cGAMP promotes the release of Glu (Figure 4D). Increased Glu release from astrocytes may be toxic to the neurons. Indeed, excess intrasynaptic and extrasynaptic Glu induces neural hyperexcitability and subsequent neuronal death, known as “glutamate excitotoxicity,” leading to neuroinflammation and neurodegenerative diseases [50]. The concentration of Glu in the CNS is strictly maintained by astrocytes due to the low threshold for neurotoxicity, approximately 0.1 to several µM, and is controlled to stay within this range, even if there are severe fluctuations due to pathological conditions [51]. The absolute amount and the tendency to continue to increase can be regarded as a trend of some abnormality that deviates from the physiological state.

There are multiple mechanisms of Glu release, including Ca^2+^-dependent exocytosis, reversal of uptake by Glu transporters, cystine/Glu antiporters, Bestrophin-1 channels, TREK-1 channels, hemichannels, and volume-regulated chloride/anion channels (VRACs) [52]. Among them, Ca^2+^-mediated exocytosis is a mechanism similar to synaptic Glu release and is considered to be the main mechanism of astrocyte Glu release under physiological conditions. The proinflammatory cytokines TNF-α and prostaglandin E2, which are induced during CNS diseases such as HIV, infections, stroke, Alzheimer’s disease, and multiple sclerosis, increase astrocytic Ca^2+^ levels and Glu exocytosis [53,54,55]. Similarly, the mechanism by which cGAMP elevates astrocyte Ca^2+^ may be mediated by TNF-α, a factor induced downstream of the STING pathway that is stimulated by cGAMP [12]. In other words, TNF-α secreted from astrocytes may increase its own Ca^2+^ concentration by autocrine action and promote Glu release.

In contrast, increased Glu release from astrocytes may be advantageous for cancer cell survival. Gliomas, primary brain tumors, have neuron-like properties and are integrated into the neural network in the brain via glutamatergic synapses, and their growth and invasion are promoted via neural excitation [56,57,58]. Regarding metastatic brain tumors, Zeng et al. reported that N-methyl-D-aspartate receptor (NMDAR)-mediated signaling, which uses Glu as a ligand, promotes breast cancer metastasis [59]. Furthermore, Zeng et al. proposed that the amount of Glu secreted by cancer cells themselves is insufficient for signaling, and that the neural gaps of glutamatergic neurons may be used as a source of Glu. In light of these reports, it has been suggested that increased Glu release from astrocytes by cGAMP may be a trigger for NMDAR signaling in cancer cells, which would contribute to tumor progression in the brain.

Another Glu-related metabolite, GABA, also seemed to be increasing in trend by cGAMP. Recent studies have reported that GABA_A_ receptor-associated protein-like 1 (GABARAPL1) promotes the growth and metastasis of breast cancer cells via metadherin (MTDH) [60]. MTDH is expressed in various types of cancers other than breast cancer cells and is known to be involved in cancer malignancy [61]. Known also as astrocyte elevated gene-1 (AEG-1) protein, which regulates EAAT2, a glutamate transporter expressed on the surface of astrocytes, it is actually involved in EAAT2 low expression in HIV-associated neurocognitive disorder (HAND) [62]. While GABA in supernatant was significantly increased by cGAMP in the present study, it is still possible that the stimulation of GABA secretion from astrocytes is only a consequence of increased Gln-Glu metabolism. Whether GABA drives downstream signaling like GABARAPL1 in cancer cells needs to be examined, but if so, it may affect tumor promotion such as NMDAR signaling.

Consistent with the present study, we confirmed that astrocytes and a human triple-negative breast cancer cell, MDA-MB-231, affect each other’s metabolome in a coculture system, and the arginine–proline pathway, which is highly glutamate-centric, was found to be the most altered metabolic pathway in astrocytes [32]. In addition to the route of GJ, cGAMP may also pass through the purine receptor P2X7R [63] and the antiporter of folate and organic phosphate (e.g., nucleotides), SLC19A1 [64]. Although their expression and function in astrocytes are unclear, intracellular uptake of cGAMP via GJ and other transport carriers may explain the metabolomic changes in coculture systems to some extent. 

In this study, we focused on glucose and Gln-Glu metabolism, which are not only the major metabolic pathways in astrocytes but also have a great influence on amino acids, nucleic acids, and lipid metabolism. Considering that metabolomic changes are the result of rapid adaptation to the environment, further investigation of the relationship with other pathways may, in turn, contribute to a full understanding of cGAMP-induced astrocytic phenotype changes.

## 4. Materials and Methods

### 4.1. Reagents

All cultures and reagents were purchased from Sigma Chemical Company (St. Louis, MO, USA), unless otherwise indicated. For the preparation of ssPalm-cGAMP and ssPalm-empty, ssPalmO-Phe (COATSOME^®^ SS-OP) was provided by NOF CORPORATION (Tokyo, Japan). Cellmatrix Type I-C (Nitta Gelatin, Osaka, Japan) was diluted with HCl (pH 3.0) to make a 0.1 mg/mL type-I collagen solution. D-glucose-^13^C6 and L-glutamic acid-^13^C5,^15^N were obtained from Taiyo Nippon Sanso (Tokyo, Japan). Ionomycin solution (Cayman Chemical, Ann Arbor, MI, USA) was diluted 100× with a mixture of ethanol and phosphate-buffered saline (PBS) (1:10) and then diluted with Hanks’ Balanced Salt solution (+) (HBSS +) to achieve the required concentrations immediately before use.

### 4.2. Preparation of ssPalm-cGAMP

cGAMP incorporating lipid nanoparticles (ssPalm-cGAMP) were prepared according to the one-pot “alcohol dilution–lyophilization method” developed by Shirane et al. [65]. The lipid solution in 90% tertiary butanol (SS-OP/DOPE/cholesterol/SUNBRIGHT GM-020 = 52.5/7.5/40/3) was prepared (1000 nmol/180 µL). An aqueous solution of cGAMP (20 µg/20 µL) in water was added to the lipid solution under vortex mixing. The mixture was diluted with 300 µL of malic acid buffer (1 mM, pH 3.0). Then, an aqueous solution of sucrose (160 mg/500 µL) was added as a cryoprotectant. The mixture was freeze-dried in a freeze-dryer (EYELA FDU-2110, as reported previously). The resulting dried samples were stored at 4 °C until used. Distilled water (500 µL) was added to freeze-dried ssPalm-cGAMP, equivalent to 20 µg of cGAMP, immediately before addition to astrocytes. Freeze-dried ssPalm-cGAMP was used within 2 days of production. and the ratio of lipid to cGAMP was lipid (nmol)/cGAMP (µg) = 50 nmoL/µg. A volume of 200 µL of ssPalm-cGAMP was used for each well of a 6-well plate (equivalent to 8 µg of cGAMP/well). 

### 4.3. Cell Culture

Astrocyte cultures were prepared from neonatal Wistar rats as described previously [66]. Briefly, the isolated cerebral cortices and hippocampi were minced and incubated with trypsin and DNase. Dissociated cells were plated in 75 cm^2^ tissue culture flasks (8–15 × 10^6^ cells/flask) precoated with 0.1 mg/mL type-I collagen solution. After 8–12 days, the cells were purified to remove fewer adherent neurons and microglia by shaking on a rotary shaker at 100 rpm for 15 h. Adherent cells were trypsinized (0.25%) and plated into 75 cm^2^ flasks. After the cells reached confluence (10 days), the confluent cells were shaken by hand for 10 min. Adherent cells were trypsinized (0.25%) and plated on new dishes. Using this method, more than 90% of the cells expressed glial fibrillary acidic protein (GFAP), a marker of astrocytes. Animal experiments were performed in accordance with the protocols approved by the Animal Research Committee of Chiba University.

The human breast cancer cell line MDA-MB-231 (MDA231) and the human lung cancer cell line, A549, were obtained from the American Type Culture Collection (ATCC, Manassas, VA, USA). The human colon cancer cell lines, DLD-1 and HCT-15, were kindly provided by Prof. Yano T at Toyo University, Japan. 

All cells were grown in Dulbecco’s modified Eagle’s medium (DMEM) (Wako Pure Chemicals, Osaka, Japan) supplemented with 10% fetal bovine serum (FBS) (Biowest, Riverside, MO, USA), penicillin (0.5 U/mL), and streptomycin (1 µg/mL) and were maintained at 37 °C in a fully humidified atmosphere of 5% CO_2_. Astrocytes were cultured with 1 mM dibutyl-cAMP as a differentiation inducer for 4 days prior to the experiment.

### 4.4. RT-PCR

Astrocytes (1 × 10^6^) (6 h after ssPalm-cGAMP or ssPalm-empty treatment) were collected by trypsin treatment. RNA extraction was performed using the RNeasy^®^ Mini Kit (Qiagen, Valencia, CA, USA) according to the manufacturer’s instructions. Reverse transcription reaction was performed using the ReverTra Ace^®^qPCR RT Master Mix (TOYOBO, Osaka, Japan) to obtain cDNA. PCR reaction solutions were prepared according to the protocol of THUNDERBIRD^®^SYBR^®^ qPCR Mix (TOYOBO), and quantitative real-time PCR reactions were performed using the StepOne™ Real-time PCR System (Applied Biosystems Japan, Tokyo, Japan). Primer information is shown in Table 1. For comparison of expression levels between groups, correction was made by dividing by the value of β-actin as an internal standard.

### 4.5. ELISA for cGAMP Detection

Astrocytes (1 × 10^6^ cells) (6 h after ssPalm-cGAMP or ssPalm-empty treatment) or cancer cell lines were scraped and then dissolved in M-PER^®^ Mammalian Protein Extraction Reagent (Thermo Fisher SCIENTIFIC K.K., Tokyo, Japan). Protein quantification was performed using the BSA method with the Pierce™ BCA Protein Assay Kit (Thermo). Doxorubicin (0.5 µM; Funakoshi, Tokyo, Japan) for the 24 h exposure group was prepared for only MDA231 to evaluate the effect of DNA damage induction. ELISA was performed according to the manufacturer’s protocol using the 2′-3′-cGAMP ELISA Kit (Cayman Chemical). Briefly, the plate was washed five times with wash buffer (200 µL/well) before use. Immunoassay Buffer C (100 µL) was added to the nonspecific binding (NSB) well, 50 µL was added to the maximum binding (B0) well, and 50 µL of each standard and sample was added to the other wells. Subsequently, 50 µL of 2′-3′-cGAMP-HRP Tracer was added to all wells, except total activity (TA) and blank (Blk), and 2′-3′-cGAMP ELISA Polyclonal Antiserum 50 µL was added to all wells, except TA, NSB, and Blk. The wells were covered with a plastic film and shaken for 2 h at room temperature. After removing the liquid from each well and washing five times with wash buffer (200 µL/well), 175 µL of TMB substrate solution was added to each well. Subsequently, 5 µL of a 1 × tracer was added to the TA well. The wells were covered with plastic film and shaken on an orbital shaker for 30 min at room temperature. Finally, 75 µL of HRP stop solution was added to each well and the absorbance at 450 nm was measured using a SpectraMax™ i3 spectrophotometer (Molecular Devices, Sunnyvale, CA, USA). The mean value of the blank was subtracted from the value of all the wells. The mean value of the NSB was subtracted from the mean value of B0 to obtain the corrected B0. Logit (B/B0) = ln [B/B0/(1 − B/B0)] for each standard and sample. The logarithm (B/B0) of the standard was plotted on the *y*-axis and the logarithm of 2′-3′-cGAMP concentration on the *x*-axis, and a calibration curve was drawn by linear approximation to calculate the sample concentration. Finally, the concentration of cGAMP in each cell was expressed as ng per mg of protein.

### 4.6. ^13^C6-Glucose Uptake

The day before the addition of cGAMP, primary cultured astrocytes were seeded in a 60 mm dish at a density of 2.2 × 10^6^ cells/4 mL/dish. The supernatant was removed and replaced with 4 mL of fresh DMEM. ssPalm-empty or ssPalm-cGAMP particle solution was added at 400 µL/dish (equivalent to 16 µg/dish of cGAMP) and incubated for 6 h. After washing with 3 mL of HBSS (+), 4 mL of HBSS (+) containing 20 mM ^13^C6-glucose was added. After 1 h of incubation, 100 μL of culture supernatant or all adherent cells were collected and fixed with methanol containing 4000-fold diluted internal standard 1 (HumanMetabolome Technologies (HMT), Yamagata, Japan). Prior to fixation, the cells were washed with 4 mL of a 5% mannitol solution. The time points for ^13^C6-glucose uptake, and the following ^13^C5,^15^N-glutamate uptake experiments were determined by examining the time at which the ^13^C substitution ratio could be confirmed in preliminary studies and dynamic changes could be observed.

### 4.7. ^13^C5,^15^N-Glutamate (Glu) Uptake

The day before the addition of cGAMP, primary cultured astrocytes were seeded in a 60 mm dish to 2.2 × 10^6^ cells/4 mL/dish. The supernatant was removed and replaced with 4 mL of fresh DMEM. ssPalm-empty or ssPalm-cGAMP particle solution was added at 400 µL/dish (equivalent to 16 µg/dish of cGAMP) and incubated for 6 h. After washing with 3 mL of HBSS (+), 4 mL of HBSS (+) containing 120 µM of ^13^C5,^15^N-glutamate (Glu) and ionomycin (0.5 µM) was added. After incubation for 5, 10, 20, and 40 min, 300 µL of supernatant or all adherent cells was collected and fixed with methanol containing 4000-fold diluted internal standard 1 (HMT). Prior to fixation, the cells were washed with 4 mL of a 5% mannitol solution.

### 4.8. Metabolite Extraction and Metabolome Analysis Using Capillary Electrophoresis Time-of-Flight Mass Spectrometry (CE-TOF-MS)

Extractions were performed as described previously [67,68]. Briefly, the detached cells were washed with 5 mL of mannitol and reconstituted in 1.0 mL of methanol containing 4000-fold diluted internal standard 1 (HMT) and then ultrasonically pulverized. In the supernatant of cell culture medium, 100 µL or 300 µL of supernatant was collected in 0.9 mL). Subsequently, 1.2 mL of CHCl_3_ and 400 µL of ultrapure water were added to the cell lysate or supernatant, and liquid–liquid extraction was performed. After centrifugation at 2300× *g* at 4 °C for 5 min, the aqueous layers were filtered through a Nanosep/3K (3-kDa cutoff) filter (Nihon Pall Ltd., Tokyo, Japan) at 9100× *g* at 4 °C for 2.5 h to remove proteins and phospholipids. The resulting filtrates were used as ionic metabolites; they were lyophilized and dissolved in 25 μL ultrapure water prior to CE-TOF-MS analysis. CE-TOF-MS was carried out using an Agilent 7100 CE system equipped with an Agilent 6230 TOF-MS System (Agilent Technologies, Santa Clara, CA, USA). Due to the characteristics of CE-TOFMS analysis where the detection time fluctuates greatly, it is desirable to run all the samples to be measured through the same measurement cycle so that the peak identification accuracy is not affected. Therefore, the number of N here indicates the number of wells (dishes) from which the samples were started. Raw data were processed using MassHunter software (Qualitative and Quantitative Analysis, Agilent Technologies) for metabolite quantification.

### 4.9. Statistical Analysis

Each experiment was performed independently at least three times or triplicates for metabolomic analysis. Data are presented as the mean ± standard error of the mean (S.E.M). Statistical analysis was performed using GraphPad Prism 5 (GraphPad Prism Software, La Jolla, CA, USA) and Student’s *t*-test for the comparison of two groups or Dunnett’s test for multiple comparison. Differences were considered statistically significant at *p* < 0.05. Metabolomic analysis was performed by Statistical Analysis and Pathway Analysis using MetaboAnalyst 5.0 (http://www.metaboanalyst.ca/ (accessed on 2 August 2021). In statistical analysis, the concentration of metabolites was converted into a Z score and analyzed. The fold changes and *p*-values for the metabolites were calculated. For the volcano plot, the *p*-values were transformed by -log10 so that the more significant features (*p*-values < 0.1 as statistically significant) could be expressed higher on the graph. In this study, we focused on both aspects of ranking: metabolic enrichment of pathways using metabolite set enrichment analysis and biological importance of metabolic pathways detected by centrality theory using topology analysis.

## 5. Conclusions

The results of our study showed that cGAMP, which is transmitted from cancer cells to astrocytes in brain tumors, may induce central metabolic alterations. cGAMP inhibits intracellular Gln synthesis and promotes the consumption of Gln in astrocytes, suggesting that it promotes the extracellular release of Glu. Future studies are needed to elucidate the effects of such metabolic changes in astrocytes on contacting metastatic cancer cells and surrounding CNS components.

## Figures and Tables

**Figure 1 ijms-22-09028-f001:**
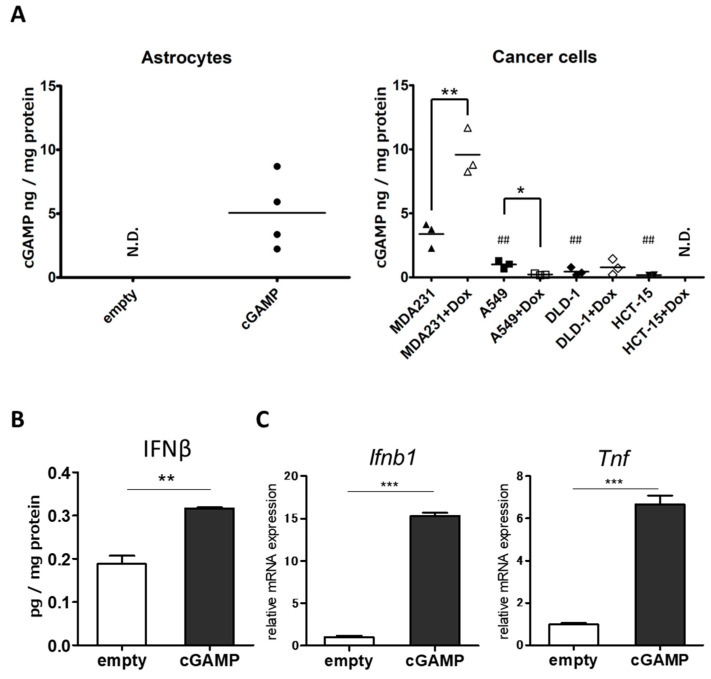
Activation of downstream STING in astrocytes by cGAMP uptake. (**A**) The amount of intracellular cGAMP in astrocytes 6 h after the addition of ssPalm (left panel) and cancer cells treated with doxorubicin for 24 h (right panel) was measured using ELISA (empty, n = 2; cGAMP, n = 4; cancer cells, n = 3). Statistical analyses were performed using Student’s *t*-test for comparison between the control and Dox treatment groups for each cancer cell (* *p* < 0.05, ** *p* < 0.01) or Dunnett’s test for comparing A549, DLD-1, and HCT-15 with MDA231 (^##^
*p* < 0.01). (**B**) The amount of IFNβ in the supernatant of astrocyte culture 6 h after the addition of ssPalm-cGAMP (n = 3, ** *p* < 0.01, Student’s *t*-test). (**C**) mRNA levels of *Ifnb1* and *Tnf* in astrocytes 6 h after the addition of ssPalm-cGAMP (n = 3, ** *p* < 0.01, *** *p* < 0.001, Student’s *t*-test).

**Figure 2 ijms-22-09028-f002:**
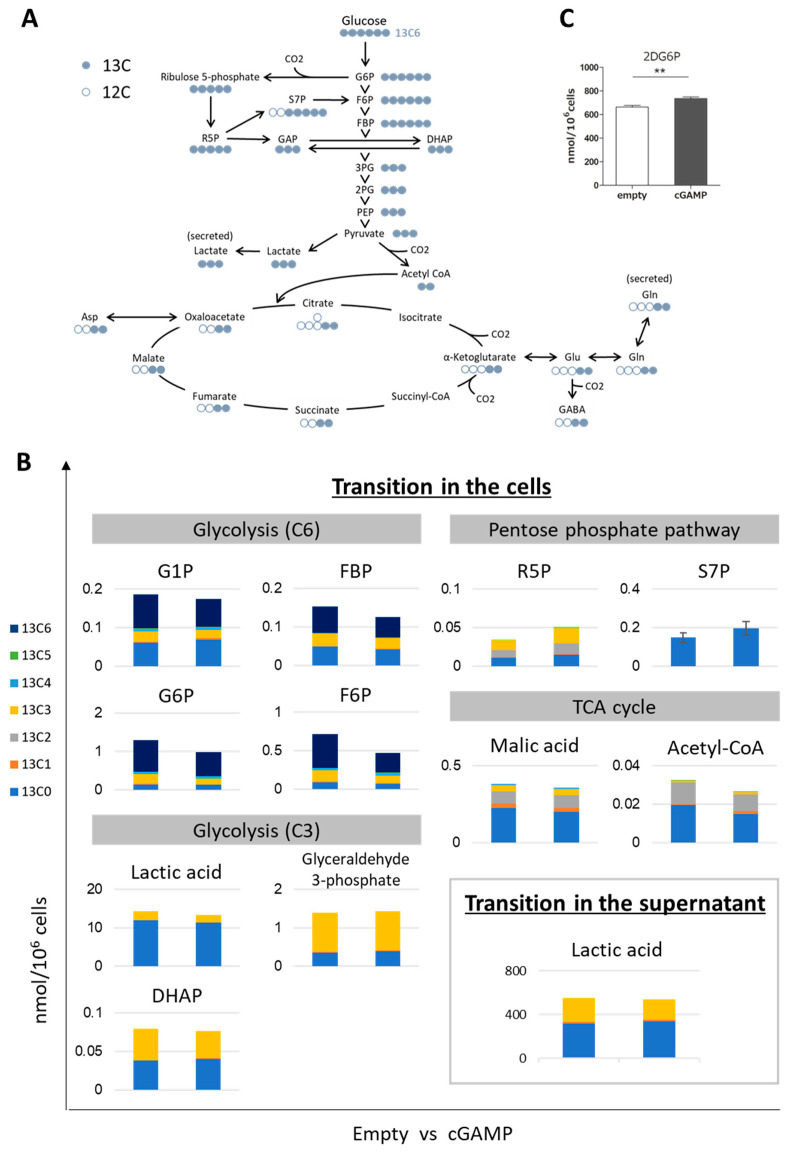
Evaluation of glucose-related metabolism in astrocytes incorporating cGAMP using ^13^C6-glucose. (**A**) Predicted distribution of ^13^C taken up as ^13^C6-glucose. For ^13^C-labeled compounds, a representation is shown of the labeled compounds that are theoretically estimated to be the most abundant when the metabolic cycle makes one turn. (**B**) The amount of each metabolite involved in the glycolytic system in cGAMP-incorporated astrocytes or their supernatant after 60 min of ^13^C6-glucose addition (n = 3). (**C**) Glucose uptake capacity was evaluated by the amount of 2DG6P in astrocytes after 60 min of 2-deoxyglucose (2-DG) addition. (n = 3, ** *p* < 0.01, Student’s *t*-test). Volcano plots (**D**) and pathway analysis (**E**) were analyzed by normalizing the concentration information of the most abundant ^13^C-labeled isotope of each metabolite measured in B. The concentration data are presented in Appendix A. R5P is the only compound with a significant increase of more than 1.5-fold vs. the empty group (*p* < 0.1, performed by MetaboAnalyst 5.0 (http://www.metaboanalyst.ca (accessed on 2 August 2021))). The top 10 pathways in decreasing order of “[-log (Raw p)] × Impact” are shown in Appendix A.

**Figure 3 ijms-22-09028-f003:**
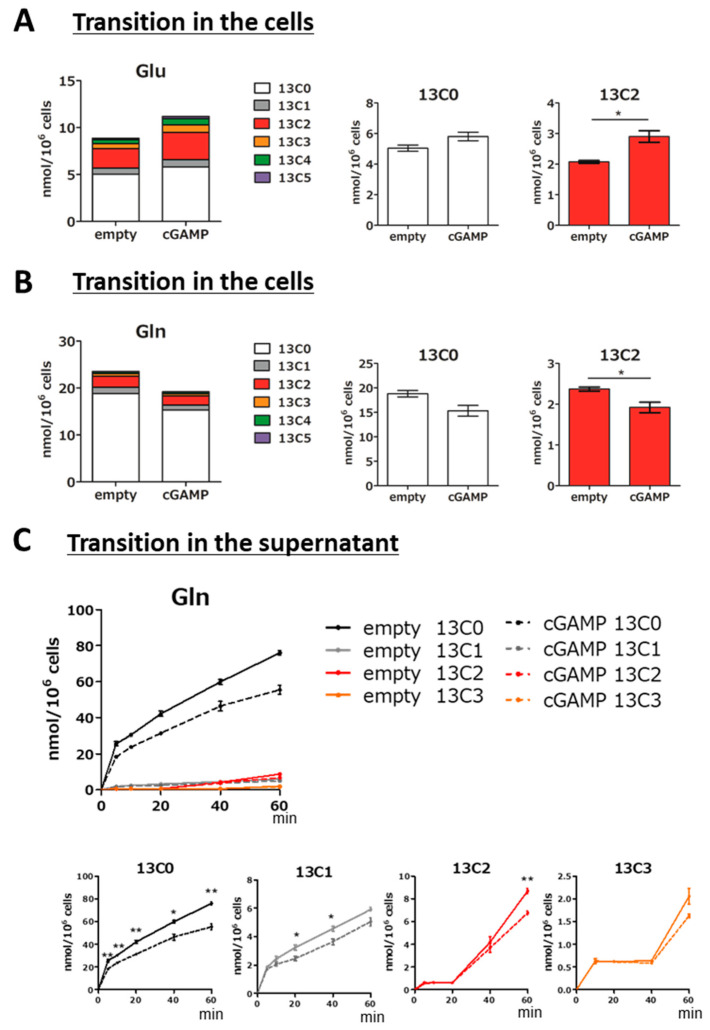
Evaluation of Glu and Gln metabolism in astrocytes incorporating cGAMP using ^13^C6-glucose. (**A**) The amount of Glu in cGAMP-incorporated astrocytes after 60 min of ^13^C6-glucose addition. (**B**) The amount of Gln in cGAMP-incorporated astrocytes after 60 min of ^13^C6-glucose addition. (**C**) Transition in the amount of Gln in the supernatant of cGAMP-incorporated astrocytes up to 60 min after ^13^C6-glucose addition. (**A**–**C**) * *p* < 0.05, ** *p* < 0.01, empty vs. cGAMP at each time point (n = 3, Student’s *t*-test).

**Figure 4 ijms-22-09028-f004:**
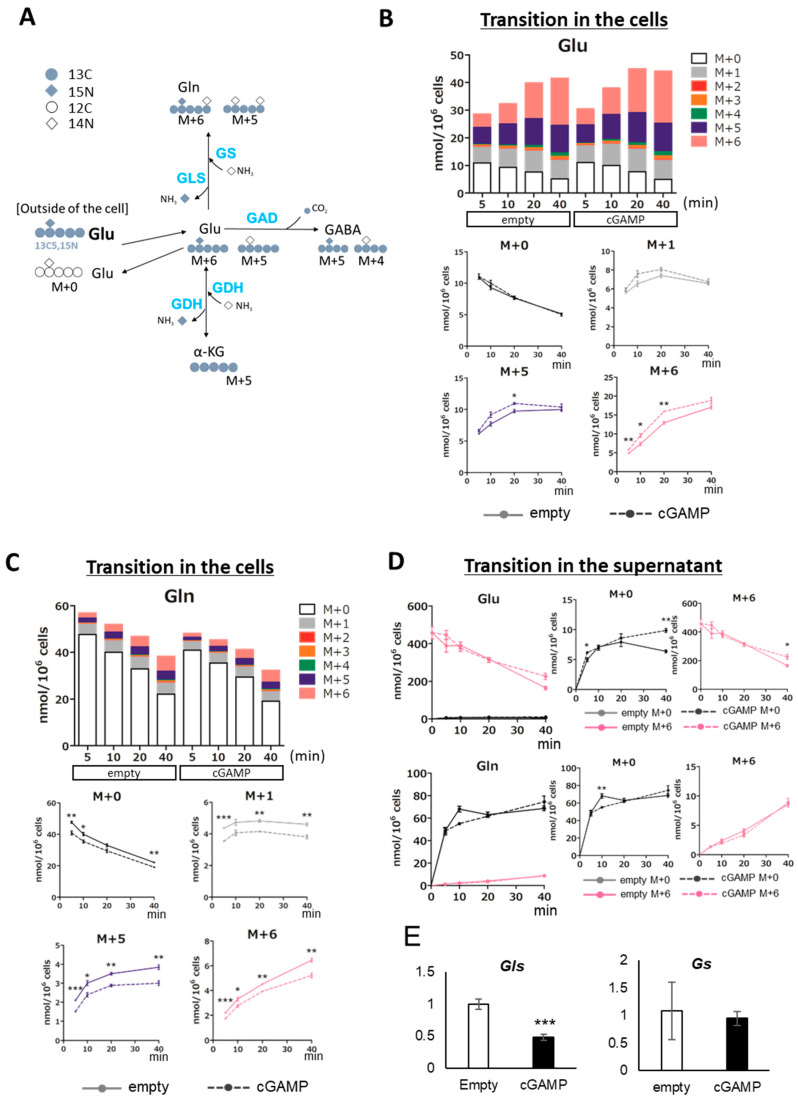
Evaluation of Glu and Gln metabolism in astrocytes incorporating cGAMP using ^13^C5,^15^N-Glu. (**A**) Predicted distribution of ^13^C and ^15^N taken up as ^13^C5,^15^N-Glu. For ^13^C or ^15^N-labeled compounds, a representation is shown of the labeled compounds that are theoretically estimated to be the most abundant when the metabolic cycle makes one turn. (**B**,**C**) Transition in the amount of Glu (**B**) and Gln (**C**) in cGAMP-incorporated astrocytes up to 40 min after 1 ^13^C5,^15^N-Glu addition. (**D**) Transition in the amount of Glu and Gln in the supernatant of cGAMP-incorporated astrocytes up to 40 min after ^13^C5,^15^N-Glu addition. (**B**–**D**) * *p* < 0.05, ** *p* < 0.01, *** *p* < 0.001, empty vs. cGAMP at each time point (n = 3, Student’s *t*-test). (**E**) Expression of *Gls* and *Gs* mRNAs was assessed by qPCR. ***; *p* < 0.001, Student’s *t*-test (n = 3).

**Figure 5 ijms-22-09028-f005:**
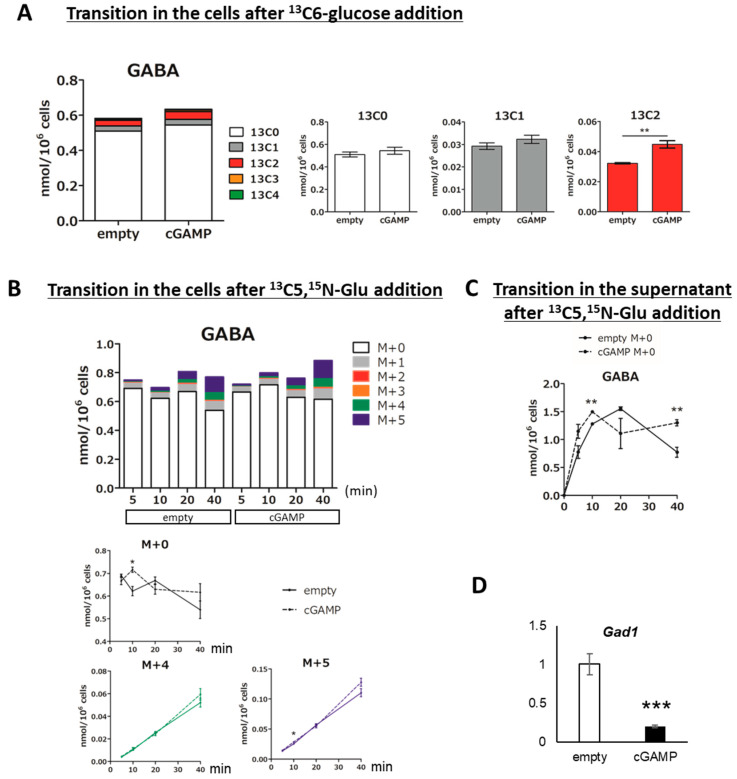
Evaluation of GABA metabolism in astrocytes incorporating cGAMP using ^13^C6-glucose and ^13^C5,^15^N-Glu. (**A**) The amount of GABA in cGAMP-incorporated astrocytes after 60 min of ^13^C6-glucose addition. (**B**) Transition in the amount of GABA in cGAMP-incorporated astrocytes up to 40 min after ^13^C5,^15^N-Glu addition. (**C**) Transition in the amount of GABA in the supernatant of cGAMP-incorporated astrocytes up to 40 min after ^13^C5,^15^N-Glu addition. (**A**–**C**) * *p* < 0.05, ** *p* < 0.01, empty vs. cGAMP at each time point (n = 3, Student’s *t*-test). (**D**) Expression of *Gad1* mRNA was assessed by qPCR. ***; *p* < 0.001, Student’s *t*-test (n = 3).

**Table 1 ijms-22-09028-t001:** RT-PCR primers.

Target Gene	Product Size	Direction	Sequences
Rat-*Actb1* (beta-actin)	104 bp	Sense	5′-CTGACAGGATGCAGAAGGAGA-3′
		Antisense	5′-AGAGCCACCAATCCACACA-3′
Rat-*Ifna1*	98 bp	Sense	5′-GGTGGTGGTGAGCTACTGGT-3′
		Antisense	5′-TTTGTGCCAGGAGTGTGAAG-3′
Rat-*Ifnb1*	89 bp	Sense	5′-TGCCCTCTCCATCGACTACA-3′
		Antisense	5′-TTCCATTCAGCTGCCTCAGG-3′
Rat-*Tnf*	96 bp	Sense	5′-ATGTGGAACTGGCAGAGGAG-3′
		Antisense	5′-CGAGCAGGAATGAGAAGAGG-3′

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
