# Peer review of "Effects on Metabolism in Astrocytes Caused by cGAMP, Which Imitates the Initial Stage of Brain Metastasis"

_ijms, 2021, doi:10.3390/ijms22169028_

Round 1

Reviewer 1 Report

  • Okawa et al. explored whether ssPalm could be used as a vehicle to insert cGAMP into astrocytes or permanent tumor cells. Additionally, in the manuscript metabolic changes after short-time stimulation with cGAMP are quantified.
  • The title is highly misleading and does not reflect the scientific approach presented in the manuscript. There are neither brain metastases nor astrocytes that have been brought to tumor cells in spatial orientation (in vitro or in vivo). Instead, it is more a question of testing an alternative (ssPalm) to the widespread established lipid-based transfections.
  • The major concerns related to this manuscript are the analysis of the own data. Only short time effects are monitored and statistical analysis are often missing.
  • The discussion does not focus on the interpretation of one's own data, but most paragraphs repeats the results or describe new technical questions that should be placed in the methods or results section.
  • The main novelty lays in the results reported in figure 4D. Here, cGAMP increases the extracellular glutamate level. However, the data are relatively weak and have to be confirmed by further measuring points.

Line107-110: Statements are not based on any static evaluations. These are mandatory.

Figure 1: The number of experiments for each group is missing. I wonder why in (A) for the cGAMP group,  only two experiments were performed.  Why several time points where included in the same diagram? I do not see any elevated cGAMP level vs. the empty group as no statistical tests were performed.

Line134: Statistics are mandatory.

Figure 2: 1 hour is a rather short time. I suggest that the data be collected after 6 hours as well. Again no statistics were performed. Please explain the legend in more detail.

Figure 3-5: Again, statistics and number of experiments are missing.

Reviewer 2 Report

In this manuscript, Okawa et al., used a special lipid nanoparticle, ssPalm to introduce cGAMP into astrocytes and studied the metabolic effect of cGAMP on astrocytes. They found that cGAMP increased glutamate and GABA production as well as astrocyte secretion, but decreased glutamine production and secretion. While the study may help to understand the effect of cGAMP on astrocytes,  how such effects may eventually have an impact on the malignant transformation of brain tumors need to be further explored. The following several questions may help to address this issue.

  1. Can the altered glucose metabolism by cGAMP in astrocyte have a tumor promoting role?
  2. Similarly, can alteration of glutamine-glutamate (Gln-Glu) metabolism by cGAMP promote the malignant transformation?
  3. Can GABA promote the malignant transformation?

Round 2

Reviewer 1 Report

Request 1-10: okay.

Reviewer 2 Report

The authors have addressed my concerns in this revision.